# A Neural PDE Solver with Temporal Stencil Modeling

## Abstract

Numerical simulation of non-linear partial differential equations plays a crucial role in modeling physical science and engineering phenomena, such as weather, climate, and aerodynamics. Recent Machine Learning (ML) models trained on low-resolution spatio-temporal signals have shown new promises in capturing important dynamics in high-resolution signals, under the condition that the models can effectively recover the missing details. However, this study shows that significant information is often lost in the low-resolution down-sampled features. To address such issues, we propose a new approach, namely Temporal Stencil Modeling (TSM), which combines the strengths of advanced time-series sequence modeling (with the HiPPO features) and state-of-the-art neural PDE solvers (with learnable stencil modeling). TSM aims to recover the lost information from the PDE trajectories and can be regarded as a temporal generalization of classic finite volume methods such as WENO. Our experimental results show that TSM achieves the new state-of-the-art simulation accuracy for 2-D incompressible Navier-Stokes turbulent flows: it significantly outperforms the previously reported best results by 19.9% in terms of the highly-correlated duration time, and reduces the inference latency into 80%. We also show a strong generalization ability of the proposed method to various out-of-distribution turbulent flow settings.

## 1 Introduction

Complex physical systems described by non-linear partial differential equations (PDEs) are ubiquitous throughout the real world, with applications ranging from design problems in aeronautics (Rhie & Chow, 1983), medicine (Sallam & Hwang, 1984), to scientific problems of molecular modeling (Lelievre & Stoltz, 2016) and astronomical simulations (Courant et al., 1967). Solving most equations of importance is usually computationally intractable with direct numerical simulations and finest features in high resolutions.

Recent advances in machine learning-accelerated PDE solvers (Bar-Sinai et al. 2019; Li et al. 2020c; Kochkov et al. 2021; Brandstetter et al. 2021, *inter alia*) have shown that end-to-end neural solvers can efficiently solve important (mostly temporal) partial differential equations. Unlike classical finite differences, finite volumes, finite elements, or pseudo-spectral methods that requires a smooth variation on the high-resolution meshes for guaranteed convergence, neural solvers do not rely on such conditions and are able to model the underlying physics with under-resolved low resolutions and produce high-quality simulation with significantly reduced computational cost.

The power of learnable PDE solvers is usually believed to come from the *super-resolution ability* of neural networks, which means that the machine learning model is capable of recovering the missing details based on the coarse features (Bar-Sinai et al., 2019; Kochkov et al., 2021). In this paper, we first empirically verify such capability by explicitly training a super-resolution model, and then find that since low-resolution down-sampling of the field can lead to some information loss, a single coarse feature map used by previous work (Kochkov et al., 2021) is not sufficient enough. We empirically show that the temporal information in the trajectories and the temporal feature encoding scheme are crucial for recovering the super-resolution details faithfully.

Motivated by the above observations, we propose Temporal Stencil Modeling (TSM), which combines the best of two worlds: stencil learning (i.e., Learned Interpolation in Kochkov et al. 2021) as that used in a state-of-the-art neural PDE solver for conservation-form PDEs, and HiPPO (Gu

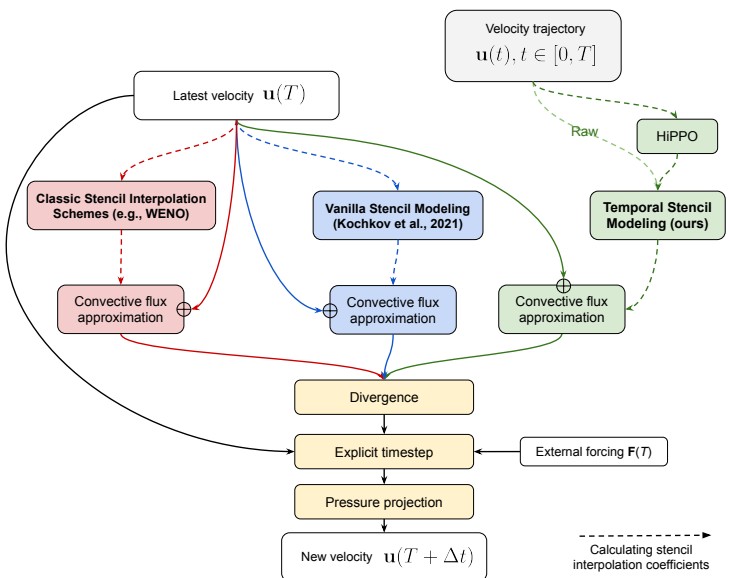

Figure 1: Illustration of classic finite volume solvers (in red color), learnable solvers with vanilla stencil modeling (in blue color) and our temporal stencil modeling (in green color). While the convective flux approximation methods are different in each method, the divergence operator, the explicit time-step operator, and the pressure projection (in yellow color) are shared between classic solvers and learnable methods. Notice that the stencil interpolation coefficients in classic solvers such as WENO can also be data-adaptive (see Sec. 3.1 for more details).

et al., 2020) as a state-of-the-art time series sequence model. Specifically, in this paper we focus on trajectory-enhanced high-quality approximation of the convective flux within a finite volume method framework. As illustrated in Fig. 1, TSM can be regarded as a **temporal generalization** of classic finite volume methods such as WENO (Liu et al., 1994; Jiang & Shu, 1996) and recently proposed learned interpolation solvers (Kochkov et al., 2021), both of which adaptively weight or interpolate the stencils based on the latest states only. On the other hand, in TSM we use the HiPPO-based temporal features to calculate the interpolation coefficients for approximating the integrated velocity on each cell surface. The HiPPO temporal features provide a good representation for calculating the interpolation coefficients, while the stencil learning framework ensures that the neural system's prediction exactly conserves the Conservation Law and the incompressibility of the fluid. With the abundant temporal information, we further utilize the temporal bundling technique (Brandstetter et al., 2021) to avoid over-fitting and improve the prediction latency for TSM.

Following the precedent work in the field (Li et al., 2020c; Kochkov et al., 2021; Brandstetter et al., 2021), we evaluate the proposed TSM neural PDE solver on 2-D incompressible Navier-Stokes equation, which is the governing equation for turbulent flows with the conservation of mass and momentum in a Newtonian fluid. Our empirical evaluation shows that TSM achieve both state-of-the-art simulation accuracy ($+19.9\%$) and inference speed ($+25\%$). We also show that TSM trained with steady-state flows can achieve strong generalization performance on out-of-distribution turbulent flows, including different forcings and different Reynolds numbers.

## 2 BACKGROUND & RELATED WORK

### 2.1 NAVIER-STOKES EQUATION

A time-dependent PDE in the conservation form can be written as

$$\partial_t \mathbf{u} + \nabla \cdot \mathbf{J}(\mathbf{u}) = 0 \tag{1}$$

where $\mathbf{u} : [0, T] \times \mathbb{X} \to \mathbb{R}^n$ is the density of the conserved quantity (i.e., the solution), $t \in [0, T]$ is the temporal dimension, $\mathbb{X} \subset \mathbb{R}^n$ is the spatial dimension, and $\mathbf{J} : \mathbb{R}^n \to \mathbb{R}^n$ is the flux, which

represents the quantity that pass or travel (whether it actually moves or not) through a surface or substance. $\nabla \cdot \mathbf{J}$ is the divergence of $\mathbf{J}$. Specifically, the incompressible, constant density 2-D Navier Stokes equation for fluids has a conservation form of:

$$\partial_t \mathbf{u} + \nabla \cdot (\mathbf{u} \otimes \mathbf{u}) = \nu \nabla^2 \mathbf{u} - \frac{1}{\rho} \nabla p + \mathbf{f} \tag{2}$$

$$\nabla \cdot \mathbf{u} = 0 \tag{3}$$

where $\otimes$ denotes the tensor product, $\nu$ is the kinematic viscosity, $\rho$ is the fluid density, $p$ is the pressure filed, and $\mathbf{f}$ is the external forcing. In Eq. 2, the left-hand side describes acceleration and convection, and the right-hand side is in effect a summation of diffusion, internal forcing source, and external forcing source. Eq. 3 enforces the incompressibility of the fluid.

A common technique to solve time-dependent PDEs is the *method of lines (MOL)* (Schiesser, 2012), where the basic idea is to replace the spatial (boundary value) derivatives in the PDE with algebraic approximations. Specifically, we discretize the spatial domain $\mathbb{X}$ into a grid $X = \mathbb{G}^n$, where $\mathbb{G}$ is a set of grids on $\mathbb{R}$. Each grid cell $g$ in $\mathbb{G}^n$ denote a small non-overlapping volume, whose center is $\mathbf{x}_g$, and the average solution value is calculated as $\mathbf{u}_g^t = \int_g \mathbf{u}(t, \mathbf{x}) d\mathbf{x}$. We then solve $\partial_t \mathbf{u}_g^t$ for $g \in \mathbb{G}^n$ and $t \in [0, T]$. Since $g \in \mathbb{G}^n$ is a set of pre-defined grid points, the only derivative operator is now in time, making it an ordinary differential equations (ODEs)-based system that approximates the original PDE.

## 2.2 CLASSICAL SOLVERS FOR COMPUTATIONAL FLUID DYNAMICS

In Computational Fluid Dynamics (CFD) (Anderson & Wendt, 1995; Pope & Pope, 2000), the Reynolds number $Re = UL/\nu$ dictates the balance between convection and diffusion, where $U$ and $L$ are the typical velocity and characteristic linear dimension. When the Reynolds number $Re \gg 1$, the fluids exhibit time-dependent chaotic behavior, known as turbulence, where the small-scale changes in the initial conditions can lead to a large difference in the outcome. The Direct Numerical Simulation (DNS) method solve Eq. 2 directly, and is a general-purpose solver with high stability. However, as $Re$ determines the smallest spatio-temporal feature scale that need to be captured by DNS, DNS faces a computational complexity as high as $O(Re^3)$ (Choi & Moin, 2012) and cannot scale to large-Reynolds number flows or large-size computation domains.

## 2.3 NEURAL PDE SOLVERS

A wide range of neural network-based solvers have recently been proposed to at the intersection of PDE solving and machine learning. We roughly classify them into four categories:

**Physics-Informed Neural Networks (PINNs)** PINN directly parameterizes the solution $\mathbf{u}$ as a neural network $F : [0, T] \times \mathbb{X} \to \mathbb{R}^n$ (Weinan & Yu, 2018; Raissi et al., 2019; Bar & Sochen, 2019; Smith et al., 2020; Wang et al., 2022). They are closely related to the classic Galerkin methods (Matthies & Keese, 2005), where the boundary condition date-fitting losses and physics-informed losses are introduced to train the neural network. These methods suffers from the parametric dependence issue, that is, for any new inital and boundary conditions, the optimization problem needs to be solved from scratch, thus limit their applications especially for time-dependent PDEs.

**Neural Operator Learning** Neural Operator methods learn the mapping from any functional parametric dependence to the solution as $F : ([0, T] \times \mathbb{X} \to \mathbb{R}^n) \to ([T, T + \Delta T] \times \mathbb{X} \to \mathbb{R}^n)$ (Lu et al., 2019; Bhattacharya et al., 2020; Patel et al., 2021). These methods are usually not bounded by fixed resolutions, and learn to directly predict any solution at time step $t \in [T, T + \Delta T]$. Fourier transform (Li et al., 2020c; Tran et al., 2021), wavelet transform (Gupta et al., 2021), random features (Nelsen & Stuart, 2021), attention mechanism (Cao, 2021), or graph neural networks(Li et al., 2020a;b) are often used in the neural network building blocks. Compared to neural methods that mimic the method of lines, the operator learning methods are not designed to generalize to dynamics for out-of-distribution $t \in [T + \Delta T, +\infty]$, and only exhibit limited accuracy for long trajectories.

**Neural Method-of-Lines Solver** Neural Method-of-Lines Solvers are autoregressive models that solve the PDE iteratively, where the difference from the latest state at time $T$ to the state at time

$T + \Delta t$ is predicted by a neural network $\mathbf{F} : ([0, T] \times X \to \mathbb{R}^n) \to (X \to \mathbb{R}^n)_{t=T+\Delta t}$. The typical choices for $\mathbf{F}$ include modeling the absolute difference: $\forall g \in X = \mathbb{G}^n$, $\mathbf{u}_g^{T+\Delta t} = \mathbf{u}_g^T + \mathbf{F}_g(\mathbf{u}_{[0,T]})$ (Wang et al., 2020; Sanchez-Gonzalez et al., 2020; Stachenfeld et al., 2021) and modeling the relative difference: $\mathbf{u}_g^{T+\Delta t} = \mathbf{u}_g^T + \Delta t \cdot \mathbf{F}_g(\mathbf{u}_{[0,T]})$ (Brandstetter et al., 2021), where the latter is believed to have the better consistency property, i.e., $\lim_{\Delta t \to 0} \|\mathbf{u}_g^{T+\Delta t} - \mathbf{u}_g^T\| = 0$.

**Hybrid Physics-ML**  Physics-ML hybrid models is a recent line of work that uses neural network to correct the errors in the classic (typically low-resolution) numerical simulators. Most of these approaches seek to learn the corrections of the numerical simulators' outputs (Mishra, 2019; Um et al., 2020; List et al., 2022; Dresdner et al., 2022; Frezat et al., 2022; Bruno et al., 2022), while Bar-Sinai et al. (2019); Kochkov et al. (2021) learn to infer the stencils of advection-diffusion problems in a Finite Volume Method (FVM) framework. The proposed Temporal Stencil Modeling (TSM) method belongs to the latter category.

## 3   TEMPORAL STENCIL MODELING FOR PDES

### 3.1   NEURAL STENCIL MODELING IN FINITE VOLUME SCHEME

Finite Volume Method (FVM) is a special MOL technique for conservation form PDEs, and can be derived from Eq. 1 via Gauss' theorem, where the integral of $\mathbf{u}$ (i.e., the averaged vector field of volume) over unit cell increases only by the net flux into the cell. Recall that the incompressible, constant density Navier Stokes equation for fluids has a conservation form of:

$$\partial_t \mathbf{u} + \nabla \cdot (\mathbf{u} \otimes \mathbf{u}) = \nu \nabla^2 \mathbf{u} - \frac{1}{\rho} \nabla p + \mathbf{f} \tag{4}$$

We can see that in FVM, the cell-average divergence can be calculated by summing the surface flux, so the problem boils down to estimating the convective flux $\mathbf{u} \otimes \mathbf{u}$ on each face. This only requires estimating $\mathbf{u}$ by interpolating the neighboring discretized velocities, called *stencils*. The beauty of FVM is that the integral of $\mathbf{u}$ is exactly conserved, and it can preserve accurate simulation as long as the flux $\mathbf{u} \otimes \mathbf{u}$ is estimated accurately. Fig. 1 illustrates an implementation (Kochkov et al., 2021) of classic FVM for the Navier-Stokes equation, where the convection and diffusion operators are based on finite-difference approximations and modeled by explicit time integration, and the pressure is implicitly modeled by the projection method (Chorin, 1967). The divergence operator enforces local conservation of momentum according to a finite volume method, and the pressure projection enforces incompressibility. The explicit time-step operator allows for the incorporation of additional time-varying forces. We refer the readers to (Kochkov et al., 2021) for more details.

Classic FVM solvers use manually designed $n^{th}$-order accurate method to calculate the interpolation coefficients of stencils to approximate the convective flux. For example, linear interpolation, upwind interpolation (Lax, 1959), and WENO5 method (Shu, 2003; Gottlieb et al., 2006) can achieve first, second, and fifth-order accuracy, respectively. Besides, adjusting the interpolation weights adaptively based on the input data is not new in numerical simulating turbulent flows. For example, given the interpolation axis, the upwind interpolation adaptively uses the value from the previous/next cell along that axis plus a correction for positive/negative velocity. WENO (weighted essentially non-oscillatory) scheme also computes the derivative estimates by taking an adaptively-weighted average over multiple estimates from different neighborhoods. However, the classic FVM solvers are designed for general cases and can only adaptively adjust the interpolation coefficients with simple patterns, and thus are sub-optimal when abundant PDE observation data is available.

In this paper, we follow Kochkov et al. (2021) and aim to learn more accurate flux approximation in the FVM framework by predicting the learnable interpolation coefficients for the stencils with neural networks. In principle, with $3 \times 3$ convolutional kernels, a $1, 2, 3$-layer neural network is able to perfectly mimic the linear interpolation, Lax-Wendroff method, and WENO method, respectively (Brandstetter et al., 2021). Such observation well connects the learnable neural stencil modeling methods to the classical schemes. However, previous work (Bar-Sinai et al., 2019; Kochkov et al., 2021) investigates the learned interpolation scheme that only adapts to the latest state $\mathbf{u}^T$, which still uses the same information as classical solvers. In this paper, we further generalize both classical and previous learnable stencil interpolation schemes by predicting the interpolation coefficients with the abundant information from all the previous trajectories $\{\mathbf{u}^t | t \in [0, T]\}$.

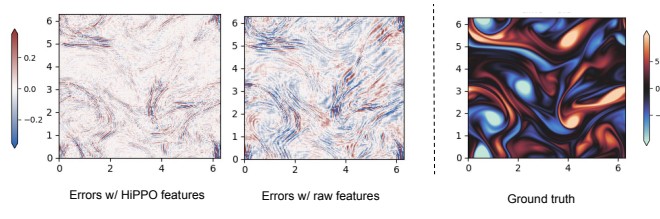

Figure 2: $64 \times 64 \rightarrow 2048 \times 2048$ super-resolution errors with $32\Delta t$-step HiPPO features and $32\Delta t$-step raw features.

Table 1: $64 \times 64 \rightarrow 2048 \times 2048$ super-resolution MSE with different approaches.

| Method | MSE |
| --- | --- |
| Bicubic Interpolation | 2.246 |
| CNN w/ 1-step raw features | 0.029 |
| CNN w/ 32-step raw features | 0.015 |
| CNN w/ 32-step HiPPO features | **0.007** |

## 3.2 Temporal Super-resolution with HiPPO features

A fundamental question in neural stencil modeling is why ML models can predict more accurate flux approximations, and previous work (Kochkov et al., 2021) attribute their success to the super-resolution power of neural nteworks, that is, the machine learning models can recover the missing details from the coarse features. In this paper, we empirically verify this hypothesis by explicitly training a super-resolution model.

Specifically, we treat the 2-D fluid velocity map as a $H \times W \times 2$ image, and train a CNN-based U-Net decoder (Ronneberger et al., 2015) to generate the super-resolution vorticity results. Our results are reported in Tab. 1 (right), and we find that the super-resolution results generated by neural networks is nearly $100 \times$ better than bicubic interpolation, which verify the super-resolution power of ML models to recover the details.

Next, since the temporal information is always available for PDE simulation[1], we investigate whether the temporal information can further reduce the super-resolution errors, i.e., recovering more details from the coarse features. After preliminary experiments, we decide to keep the convolutional neural networks as the spatial encoder module for their efficient implementation on GPUs and translation invariant property, and only change the temporal input features. Inspired by the recent progress in time series modeling, we consider the following two types of features as the CNN model's inputs:

**Raw features** Following the previous work on video super-resolution (Liao et al., 2015), we treat the $H \times W \times T \times 2$ -shaped velocity trajectory as an $H \times W$ image with feature channels $C = T \times 2$.

**HiPPO features** HiPPO (High-order Polynomial Projection Operators) (Gu et al., 2020; 2021) is a recently proposed framework for the online compression of continuous time series by projection onto polynomial bases. It computes the optimal polynomial coefficients for the scaled Legendre polynomial basis. It has been shown as a state-of-the-art autoregressive sequence model for raw images (Tay et al., 2020) and audio (Goel et al., 2022). In this paper, we propose to adopt HiPPO to encode the raw fluid velocity trajectories. Due to the space limit, we leave the general description of the HiPPO technique to Appendix C.

**Results** We report the temporal super-resolution results in Tab. 1. From the table, we can see that the $32\Delta t$-step raw features can reduce the super-resolution error by half, and using the HiPPO to encode the time series can further reduce the error scale by half. We also visualize the errors of $32\Delta t$-step HiPPO features and $32\Delta t$-step raw features in Fig. 2. Our temporal super-resolution results show that plenty of information in the PDE low-resolution temporal trajectories is missed with vanilla stencil modeling, or will be underutilized with raw features, and HiPPO can better exploit the temporal information in the velocity trajectories.

## 3.3 Temporal Stencil Modeling with HiPPO Features

As better super-resolution performance indicate that more details can be recovered from the low-resolution features, we propose to compute the interpolation coefficients in convective flux with

---

[1]Even if we only have access to one step of the ground-truth trajectory, we can still use the high-resolution DNS to generate a long enough trajectory to initialize the spatial-temporal model.

the HiPPO-based temporal information, which should lead to more accurate flux approximation. Incorporating HiPPO features in the neural stencil modeling framework is straight-forward: with ground-truth initial velocity trajectories $\mathbf{v}_{[0,T]}$, we first recurrently encode the trajectory (step by step with Eq. 7 in Appendix C), and use the resulted features $\text{HiPPO}(\mathbf{v}_{[0,T]})$ to compute the interpolation coefficients for the stencils. Given model-generated new velocity $\mathbf{v}_{T+\Delta t}$, due to the recurrence property of HiPPO, we can only apply a single update on $\text{HiPPO}(\mathbf{v}_{[0,T]})$ and get the new encoded feature $\text{HiPPO}(\mathbf{v}_{[0,T+\Delta t]})$, which is very efficient. Fig. 9 in the appendix illustrates such process.

# 4 EXPERIMENTS

## 4.1 EXPERIMENTAL SETUP

**Simulated data** Following previous work (Kochkov et al., 2021), we train our method with 2-D Kolmogorov flow, a variant of incompressible Navier-Stokes flow with constant forcing $\mathbf{f} = \sin(4y)\hat{\mathbf{x}} - 0.1\mathbf{u}$. All training and evaluation data are generated with a JAX-based[2] finite volume-based direct numerical simulator in a staggered-square mesh (McDonough, 2007) as briefly described in Sec. 3.1. We refer the readers to the appendix of (Kochkov et al., 2021) for more data generation details.

We train the neural models on $Re = 1000$ flow data with density $\rho = 1$ and viscosity $\nu = 0.001$ on a $2\pi \times 2\pi$ domain, which results in a time-step of $\Delta t = 7.0125 \times 10^{-3}$ according to the Courant–Friedrichs–Lewy (CFD) condition on the $64 \times 64$ simulation grid. For training, we generate 128 trajectories of fluid dynamics, each starts with different random initial conditions and simulates with $2048 \times 2048$ resolution for $40.0$ time units. We use 16 trajectories for evaluation.

**Unrolled training** All the learnable solvers are trained with the Mean Squared Error (MSE) loss on the velocities. Following previous work (Li et al., 2020c; Brandstetter et al., 2021; Kochkov et al., 2021; Dresdner et al., 2022), we adopt the unrolled training technique, which requires the learnable solvers to mimic the ground-truth solution for more than one unrolled decoding step:

$$\mathcal{L}(\mathbf{u}_{[0,T]}^{gt}) = \frac{1}{N} \sum_{i=1}^{N} \text{MSE}(\mathbf{u}^{gt}(t_i), \mathbf{u}^{pred}(t_i)) \tag{5}$$

where $t_i \in \{T + \Delta t, \ldots, T + N\Delta t\}$ is the unrolled time steps, $\mathbf{u}_{gt}$ and $\mathbf{u}_{pred}$ are the ground-truth solution and learnable solver's prediction, respectively. Unrolled training can improve the inference performance but makes the training less stable (due to bad initial predictions), so we use $N = 32$ unrolling steps for training.

**Temporal bundling** In our preliminary experiments, we found that due to the abundant information in the trajectories, the TSM solvers are more prone to over-fit. Therefore, we adopt the temporal bundling technique (Brandstetter et al., 2021) to the learned interpolation coefficients in TSM solvers. Assume that in step-by-step prediction scheme, we predict $\mathbf{u}_{[0,T]} \rightarrow \mathbf{c}(T + \Delta t)$, where $\mathbf{c}$ is the stencil interpolation coefficients in the convective flux approximation. In temporal bundling, we predict $K$ steps of the interpolation coefficients $\mathbf{u}_{[0,T]} \rightarrow \{\mathbf{c}(T+\Delta t), \mathbf{c}(T+2\Delta t), \ldots, \mathbf{c}(T+K\Delta t)\}$ in advance, and then time-step forward the FVM physics model for $K$ steps with pre-computed stencil interpolation coefficients.

**Neural network architectures & hyper-parameters** In TSM-$64 \times 64$ with a $T$-length trajectory, the input and output shapes of TSM are $64 \times 64 \times T \times 2$ and $64 \times 64 \times C \times 2$, while the input and output shapes of CNN are $64 \times 64 \times (C \times 2)$ and $64 \times 64 \times (8 \times (4^2 - 1))$, which represents that for each resolution, we predict 8 interpolations that need 15 inputs each. For HiPPO[3], we set the hyper-parameters $a = -0.5, b = 1.0, dt = 1.0$. For CNN, we use a 6-layer network with $3 \times 3$ kernels and 256 channels with periodic padding[4].

---

[2]https://github.com/google/jax-cfd
[3]https://github.com/HazyResearch/state-spaces
[4]https://github.com/google/jax-cfd/blob/main/jax_cfd/ml/towers.py

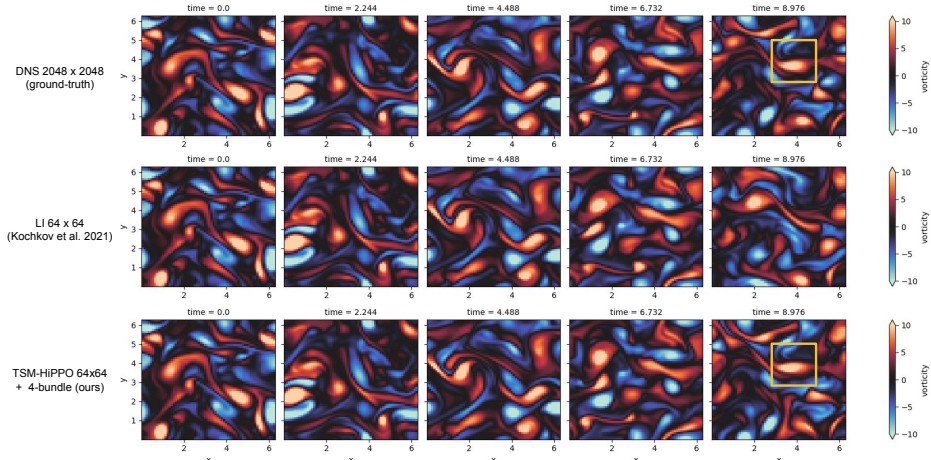

Figure 3: Qualitative results of predicted vorticity fields for reference (DNS $2048 \times 2048$), previous learnable sota model (LI $64 \times 64$) (Kochkov et al., 2021), and our method (TSM $64 \times 64$), starting from the same initial condition. The yellow box denotes a vortex that is not captured by LI.

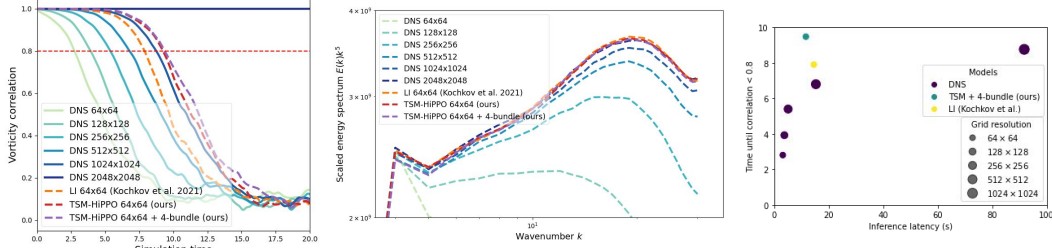

Figure 4: (left) Comparison of the vorticity correlation between prediction and the ground-truth solution (i.e., DNS $2048 \times 2048$). (middle) Energy spectrum scaled by $k^5$ averaged between simulation time 6.0 to 20.0. (right) Comparison of high vorticity correlation duration v.s. inference latency.

## 4.2 MAIN RESULTS

The classic and neural methods we evaluated can be roughly classified into four categories: 1) pure Physics models, i.e., the FVM-based Direct Numerical Simulation (DNS), 2) pure Machine Learning (ML)-based neural method-of-lines models, 3) Learned Correction (LC) models, which correct the final outputs (i.e., velocities) of physics models with neural networks (Um et al., 2020), and 4) Neural Stencil Modeling models, including single-time-step Learned Interpolation (LI) (Kochkov et al., 2021), and our Temporal Stencil Modeling (TSM).

For neural network baselines, except the periodic[5] Convolutional Neural Networks (CNN) (LeCun et al., 1999; Kochkov et al., 2021) with raw and HiPPO features we already described, we also compare with Fourier Neural Operators (FNO) (Li et al., 2020c) and Multiwavelet-based model (MWT), two recent state-of-the-art pure-ML PDE solver based on spectral and wavelet features.

We evaluate all the solvers based on their Pearson correlation $\rho$ with the ground-truth (i.e., DNS with the highest $2048 \times 2048$ resolution) flows in terms of the scalar vorticity field $\omega = \partial_x u_y - \partial_y u_x$. Furthermore, to ease comparing all the different solvers quantitatively, we focus on their high-correlation duration time, i.e., the duration of time until correlation $\rho$ drops below 0.8.

The comparison between HiPPO-based TSM and 1-time-step raw-feature LI (Kochkov et al., 2021) is shown in Fig. 3 and Fig. 4 (left). We can see that our HiPPO feature-based TSM significantly outperforms the previous state-of-the-art ML-physics model, especially when trained with a 4-step temporal bundling. From Fig. 4 (middle), we can see that all the learnable solvers can better capture the high-frequency features with a similar energy spectrum $E(\mathbf{k}) = \frac{1}{2}|\mathbf{u}(\mathbf{k})|^2$ pattern as the high-resolution ground-truth trajectories. From Fig. 4 (right), we can see that with the help of temporal

---

[5]https://github.com/google/jax-cfd/blob/main/jax_cfd/ml/layers.py

Table 2: Quantitative comparisons with the metric of high-correlation ($\rho > 0.8$) duration (w.r.t the reference DNS-2048 × 2048 trajectories). All learnable solvers use the 64 × 64 grids.

| Method | Type | High-corr. duration |
|---|---|---|
| DNS-64 × 64 | Physics | 2.805 |
| DNS-128 × 128 | Physics | 3.983 |
| DNS-256 × 256 | Physics | 5.386 |
| DNS-512 × 512 | Physics | 6.788 |
| DNS-1024 × 1024 | Physics | 8.752 |
| 1-step-raw-CNN | ML | 4.824 |
| 4-step-raw-CNN | ML | 7.517 |
| 32-step-FNO | ML | 6.283 |
| 32-step-WMT | ML | 5.890 |
| 1-step-raw-CNN | LC | 6.900 |
| 32-step-FNO | LC | 7.630 |
| 1-step-raw-CNN | LI | 7.910 |
| 32-step-FNO | TSM | 7.798 |
| 4-step-raw-CNN | TSM | 8.359 |
| + 4-step temporal-bundle | TSM | 8.303 |
| 32-step-HiPPO-CNN | TSM | 9.256 |
| + 4-step temporal-bundle | TSM | **9.481** |

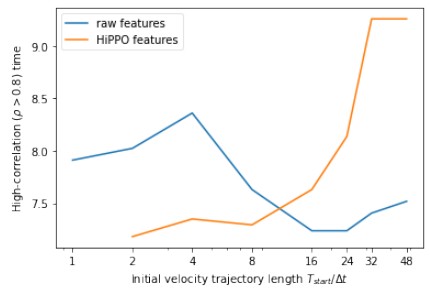

Figure 5: Temporal stencil modeling performance (high-correlation duration) with different feature types and different initial trajectory steps.

bundling, HiPPO-TSM can achieve high simulation accuracy while reduces the inference latency into 80% when compared to original LI.

### 4.3 ABLATION STUDY

We present a quantitative comparison for more methods in Tab. 1. We can see that the learned correction models are always better than pure-ML models, and neural stencil modeling models are always better than learned correction models. In terms of neural network architectures, we find that under the same ML-physics framework, raw-feature CNN is always better than FNO, and HiPPO-feature CNNs are always better than raw-feature CNNs. Finally, when adopting temporal bundling, we can see that only HiPPO-TSM can benefit from alleviating the over-fitting problem, while the performance of raw-feature TSM can be hurted by temporal bundling.

We also study the impact of the trajectory length on raw-feature and HiPPO-feature TSM models in Fig. 2. Notice that in raw features, the temporal window size is fixed during unrolling, while in HiPPO features, the temporal window size is expanded during unrolling decoding. From the figure, we can see that the raw-feature CNN achieves the best performance with a window size of 4, while HiPPO-feature keep increasing the performance, and reach the peak with 32 initial trajectory length.

### 4.4 GENERALIZATION TESTS

We evaluate the generalization ability of our HiPPO-TSM (4-step bundle) model and LI trained on trained on Kolmogorov flows ($Re = 1000$). Specifically, we consider the following test cases: (A) decaying flows (starting $Re = 1000$), (B) more turbulent Kolmogorov flows ($Re = 4000$), and (C) 2× larger domain Kolmogorov flows ($Re = 1000$). Our results are shown in Fig. 6, from which we can see that HiPPO-TSM achieves consistent improvement over LI. HiPPO-TSM also achieves competitive performance to DNS-1024 × 1024 or DNS-2048 × 2048, depending on the ground-truth (i.e., highest resolution) being DNS-2048 × 2048 or DNS-4096 × 4096.

## 5 CONCLUSION & FUTURE WORK

In this paper, we propose a novel Temporal Stencil Modeling (TSM) method for solving time-dependent PDEs in conservation form. TSM can be regarded as the temporal generalization of classic finite volume solvers such as WENO and vanilla neural stencil modeling methods (Kochkov et al., 2021), in that TSM leverages the temporal information from trajectories, instead of only using

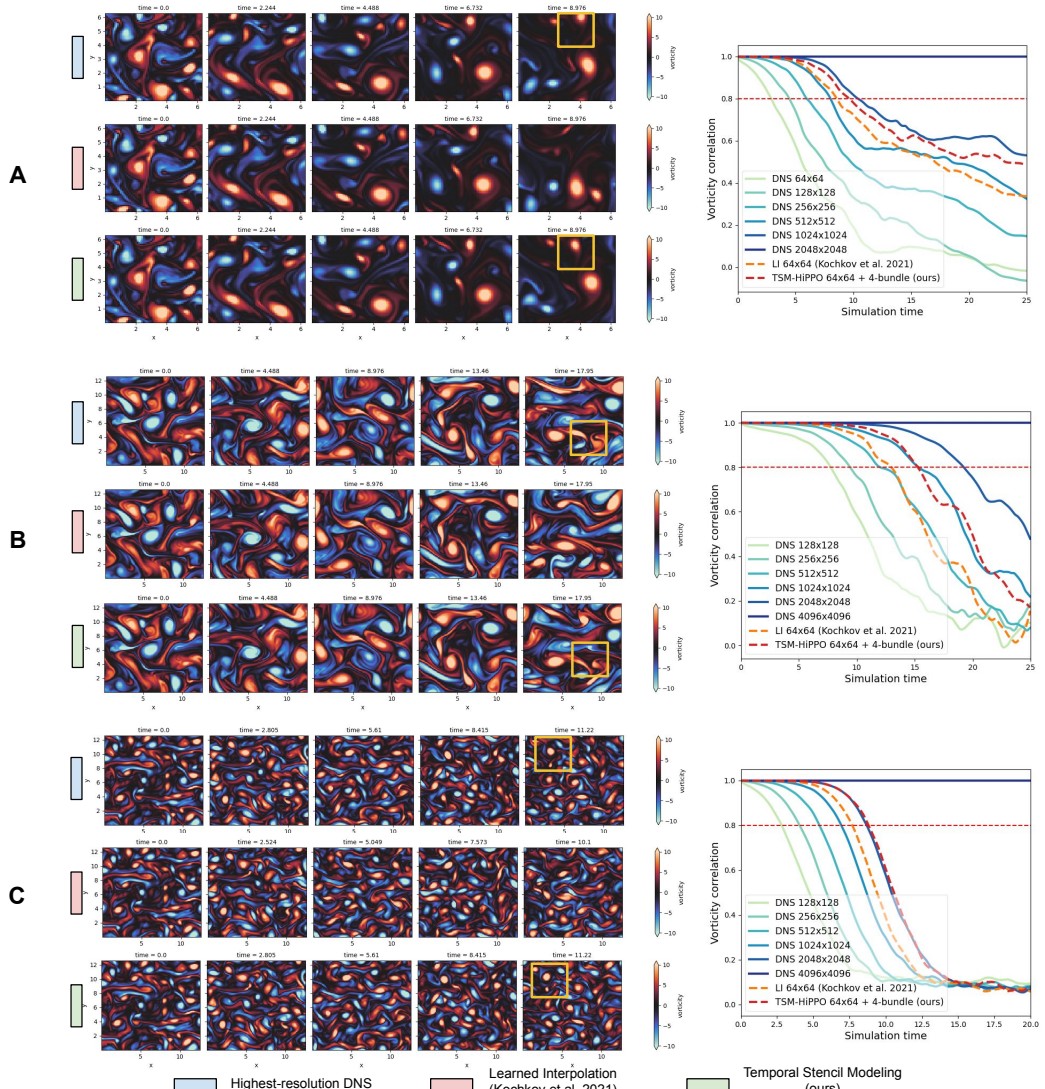

Figure 6: Generalization test results of neural methods trained on Kolmogorov flows ($Re = 1000$) and evaluated on (A) decaying flows (starting from $Re = 1000$), (B) more turbulent Kolmogorov flows ($Re = 4000$), and (C) $2\times$ larger domain Kolmogorov flows ($Re = 1000$).

the latest states, to approximate the (convective) flux more accurately. Our empirical evaluation on 2-D incompressible Navier-Stokes turbulent flow data show that both the temporal information and its temporal feature encoding scheme are crucial to achieve state-of-the-art simulation accuracy. We also show that TSM have strong generalization ability on various out-of-distribution turbulent flows.

For future work, we plan to evaluate our TSM method on the more challenging and realistic 3-D turbulent flows, as well as 2-D turbulent flows with non-periodic boundary conditions. We are also interested in leveraging the Neural Architecture Search (NAS) technique to automatically find better features and neural architectures for solving Navier-Stokes equation in the TSM framework.

## ETHIC STATEMENT

We do not see obvious negative social impact of our work.

## REPRODUCIBILITY STATEMENT

We release the source code at `https://anonymous-url`. We plan to open-source all the code after the acceptance of the manuscript. Since the training and evaluation data in this paper are all simulated, they can also be faithfully reproduced from our release code.

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

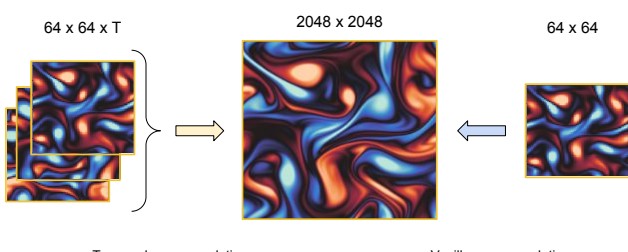

Figure 7: Illustration of super-resolution process from a trajectory of frames (i.e., temporal super-resolution) or a single-frame (vanilla super-resolution).

## A ADDITIONAL RELATED WORK

### A.1 INDUSTRIAL SOLVERS FOR COMPUTATIONAL FLUID DYNAMICS

Industrial CFD typically relies on either Reynolds-averaged Navier-Stokes (RANS) models (Boussinesq, 1877; Alfonsi, 2009), where the fluctuations are expressed as a function of the eddy viscosity, or coarsely-resolved Large-Eddy Simulation (LES) (Smagorinsky, 1963; Lesieur & Metais, 1996), where only the large scales are numerically simulated, and the small ones are modelled (with an a-priori physics assumption). However, there are severe limits in both methods associated with their usage for general purpose. For example, RANS is too simple to model complex flows (Slotnick et al., 2014) and often exhibit significant errors when dealing with complex pressure-gradient distributions and complicated geometries. LES can exhibit limited accuracy in their predictions of high-$Re$ turbulent flows, due to the first-order dependence of LES on the subgrid-scale (SGS) model.

## B INFRASTRUCTURES

We train and evaluate all the classic and neural Navier-Stokes solvers in 8 Nvidia Tesla V100-32G GPUs. The inference latency is measured by unrolling 2 trajectories for 25.0 simulation time on a single V100 GPU.

## C HIPPO FEATURES FOR TEMPORAL STENCIL MODELING

For scalar time series $u_{\leq t} := u(x)|_{x \leq t}$, the HiPPO (Gu et al., 2020) projection aims to find a coefficient mapping for orthogonal polynomials such that

$$c(t) \in \mathbb{R}^N = \arg\min \|u_{\leq t} - g^{(t)}\|_{\mu(t)} \tag{6}$$

where $g^{(t)} = \sum_{n=1}^{N} c_n(t) g_n(t)$ and $g_n(t)$ is the $n^{th}$ Legendre polynomial scaled to the $[0, t]$ domain. $\mu(t) = \frac{1}{t}\mathbb{I}_{[0,t]}$ is the uniform weight metric. By solving the corresponding ODE and its discretization, Gu et al. (2020) showed that the optimal polynomial coefficients $c_T$ can be calculated by the following recurrence:

$$c(T + \Delta t) = \left(1 - \frac{A}{T/\Delta t}\right) c(T) + \frac{1}{T/\Delta t} B \cdot u(T) \tag{7}$$

where
$$A_{nk} = \begin{cases} (2n+1)^{1/2}(2k+1)^{1/2} & \text{if } n > k \\ n+1 & \text{if } n = k \ , \\ 0 & \text{if } n < k \end{cases} \qquad B_n = (2n+1)^{\frac{1}{2}}$$

We refer the readers to (Gu et al., 2020) for the detailed derivations. When dealing with the multivariate time series such as temporal PDE, we follow the original recipe and treat each scalar component independently, that is, we treat the $H \times W \times T \times 2$ -shaped velocity trajectory as $H \times W \times 2$ separate time series of length $T$. Finally, we concatenate the resulted $H \times W \times 2 \times C$ features on the velocity dimension (i.e., $H \times W \times 2C$) as the input to the CNN.

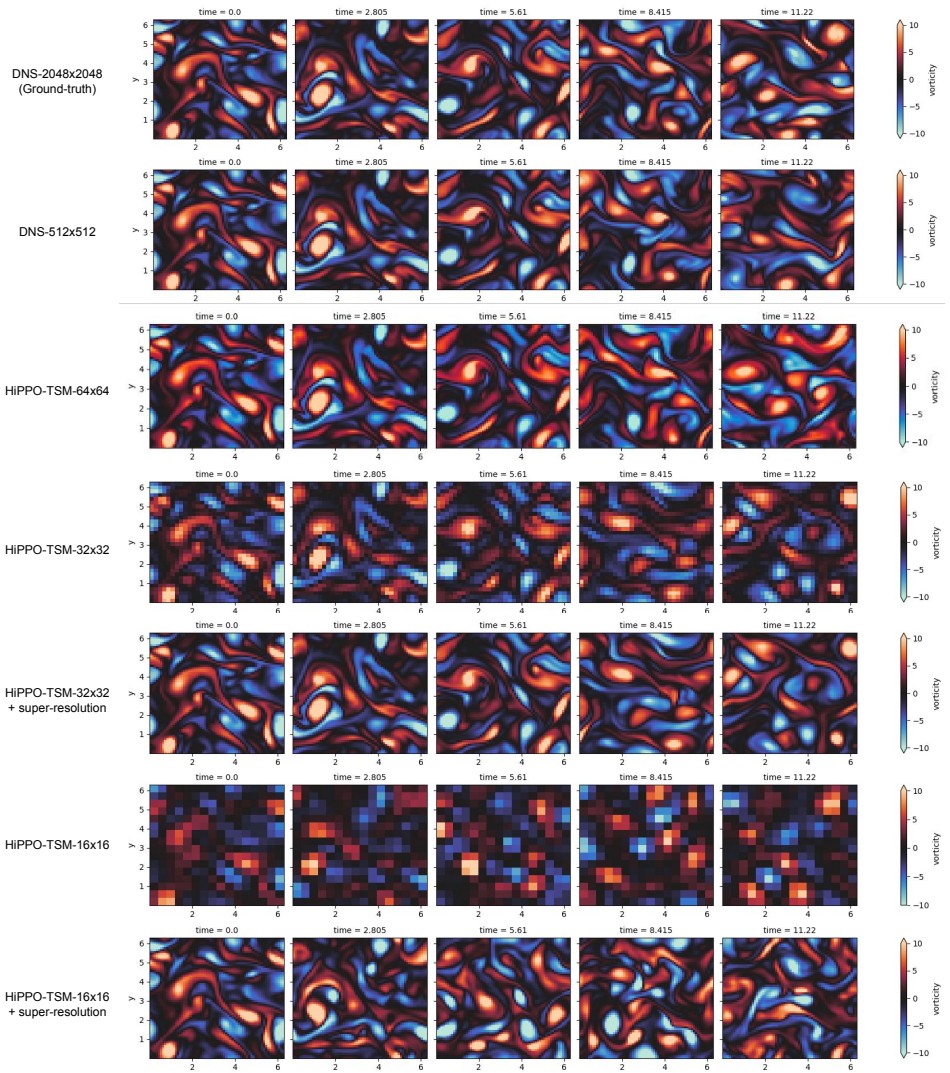

Figure 8: Qualitative results with TSM evaluating on $32 \times 32$ and $16 \times 16$, and their $64 \times 64$ super-resolution results. From the super-resolution results of the $T = 0$, we can see that the super-resolution models actually work quite well.

## D  SOLVING NAVIER-STOKES WITH $32 \times 32$ AND $16 \times 16$ GRIDS

Since TSM shows significant improvements over vanilla neural stencil modeling on the $64 \times 64$ grid, we are wondering whether TSM can further push the Pareto frontier by accurately approximate the convective flux in lower resolution grid.

Specifically, we still use DNS-$2048 \times 2048$ as the ground-truth training data, but train the TSM solver in the $32 \times 32$ and $16 \times 16$ down-sampled grids. In order to make a fair and direction comparison to our original $64 \times 64$ model, we additionally train two $32 \times 32 \rightarrow 64 \times 64$ and $16 \times 16 \rightarrow 64 \times 64$ super-resolution model, and evaluate (on $64 \times 64$ vorticity) their super-resolved results.

We report the results in Fig. 8 and Tab. 3. We can see that the super-resolution models actually work quite well for $32 \times 32 \rightarrow 64 \times 64$ and $16 \times 16 \rightarrow 64 \times 64$. Besides, though the lower-resolution neural solvers' performance is significantly worse than $64 \times 64$, we can see that the improvement from temporal information in the trajectories is more significant in lower resolutions.

Table 3: Quantitative comparisons with the metric of high-correlation ($\rho > 0.8$) duration (w.r.t the reference DNS-2048 $\times$ 2048 trajectories). The results of TSM-32 $\times$ 32 and TSM-16 $\times$ 16 are evaluated on their corresponding 64 $\times$ 64 super-resolution results with additionally trained super-resolution models.

| Method | High-corr. duration |
|---|---|
| DNS-64 $\times$ 64 | 2.805 |
| DNS-128 $\times$ 128 | 3.983 |
| DNS-256 $\times$ 256 | 5.386 |
| DNS-512 $\times$ 512 | 6.788 |
| DNS-1024 $\times$ 1024 | 8.752 |
| LI-64 $\times$ 64 | 7.910 |
| TSM-64 $\times$ 64 | **9.481 (+19.86%)** |
| LI-32 $\times$ 32 | 5.400 |
| TSM-32 $\times$ 32 | **6.802 (+25.96%)** |
| LI-16 $\times$ 16 | 2.805 |
| TSM-16 $\times$ 16 | **3.576 (+27.50%)** |

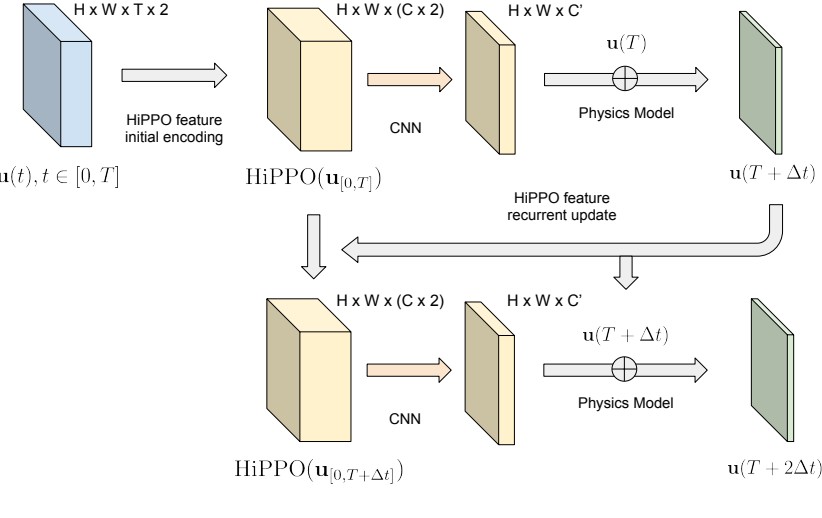

...

Figure 9: Illustration of using HiPPO features as the CNN inputs for neural stencil modeling. After encoding the HiPPO features for the initial velocity trajectory $\mathbf{v}_{[0,T]}$, given model-generated new velocity, we only need an additional recurrent update step to get the HiPPO features for $\mathbf{v}_{[0,T+\Delta t]}$ (with Eq. 7). Notice that in TSM, we use the fixed optimal HiPPO recurrence for temporal encoding and only the CNN part is learnable.

Table 4: The $p$-values in one-sample T-test for the differences between TSM and other baseline models. The differences on all 16 test trajectories are used for each significance test. We evaluate the significance for four high-correlation thresholds: 0.95, 0.9, 0.8, and 0.7.

| Baseline | $p$-value in one-sample T-test | | | |
| | $\rho > 0.95$ | $\rho > 0.9$ | $\rho > 0.8$ | $\rho > 0.7$ |
|---|---|---|---|---|
| DNS 64x64 | $1.06 \times 10^{-11}$ | $1.14 \times 10^{-10}$ | $1.20 \times 10^{-10}$ | $1.26 \times 10^{-10}$ |
| DNS 128x128 | $8.80 \times 10^{-11}$ | $2.23 \times 10^{-10}$ | $6.75 \times 10^{-10}$ | $6.63 \times 10^{-10}$ |
| DNS 256x256 | $1.22 \times 10^{-10}$ | $2.39 \times 10^{-09}$ | $4.78 \times 10^{-09}$ | $1.56 \times 10^{-09}$ |
| DNS 512x512 | $2.38 \times 10^{-09}$ | $1.49 \times 10^{-07}$ | $2.27 \times 10^{-06}$ | $1.65 \times 10^{-06}$ |
| DNS 1024x1024 | $6.63 \times 10^{-03}$ | $6.91 \times 10^{-03}$ | $1.53 \times 10^{-02}$ | $1.65 \times 10^{-02}$ |
| LI 64x64 (Kochkov et al. 2021) | $9.06 \times 10^{-04}$ | $1.65 \times 10^{-03}$ | $3.63 \times 10^{-03}$ | $3.08 \times 10^{-03}$ |

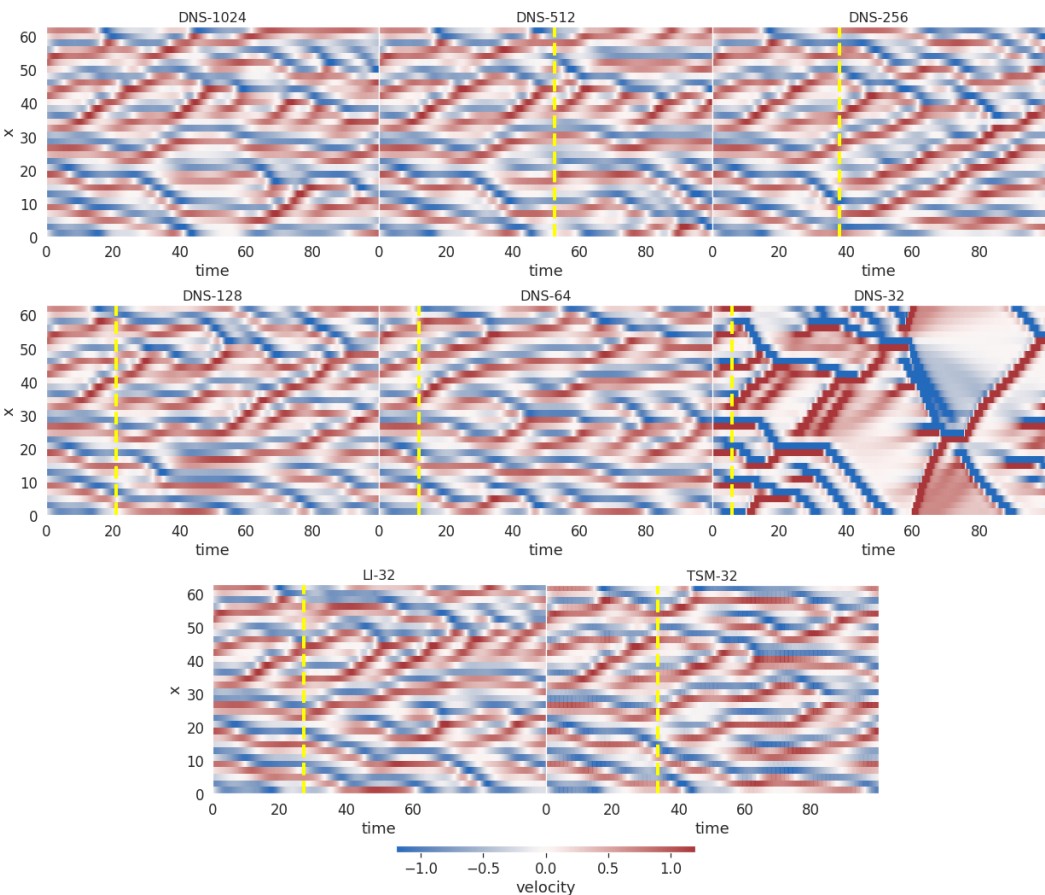

Figure 10: Qualitative comparison of TSM and other baselines on 1D KS equation. The solutions are down-sampled to 32 grid in the space dimension for comparison. The dashed vertical yellow line denotes the time-step where the Pearson correlation between the model prediction and ground-truth (i.e., DNS-1024) is lower than threshold ($\rho < 0.8$).

## E  GENERALIZATION ON 1D KURAMOTO–SIVASHINSKY (KS) EQUATION

To verify the generalization ability of TSM, we further evaluate it on 1D equations. Following previous work (Bar-Sinai et al., 2019; Stachenfeld et al., 2021), we choose Kuramoto–Sivashinsky (KS) equation as a representative 1D PDE that can generate unstable and chaotic dynamics. While KS-1D is not technically turbulent, it is a well-studied chaotic equation that can be used to assess the generalization ability of our models in 1D cases.

Specifically, the KS equation can be written in the conservation form of

$$\frac{\partial v}{\partial t} + \frac{\partial J}{\partial x} = 0, \qquad v(x, t = 0) = v_0(x) \tag{8}$$

where

$$J = \frac{v^2}{2} + \frac{\partial v}{\partial x} + \frac{\partial^3 v}{\partial x^3} \tag{9}$$

Following previous work (Bar-Sinai et al., 2019), we consider the 1D KS equation with periodic boundaries. The domain size is set to $L = 20\pi$, and the initial condition is set to

$$v_0(x) = \sum_{i=1}^{N} A_i \sin\left(2\pi \ell_i x / L + \phi_i\right) \tag{10}$$

where $N = 10$, and $A$, $\phi$, $\ell$ are sampled from the uniform distributions of $[-0.5, 0.5]$, $[-\pi, \pi]$, and $\{1, 2, 3\}$, respectively.

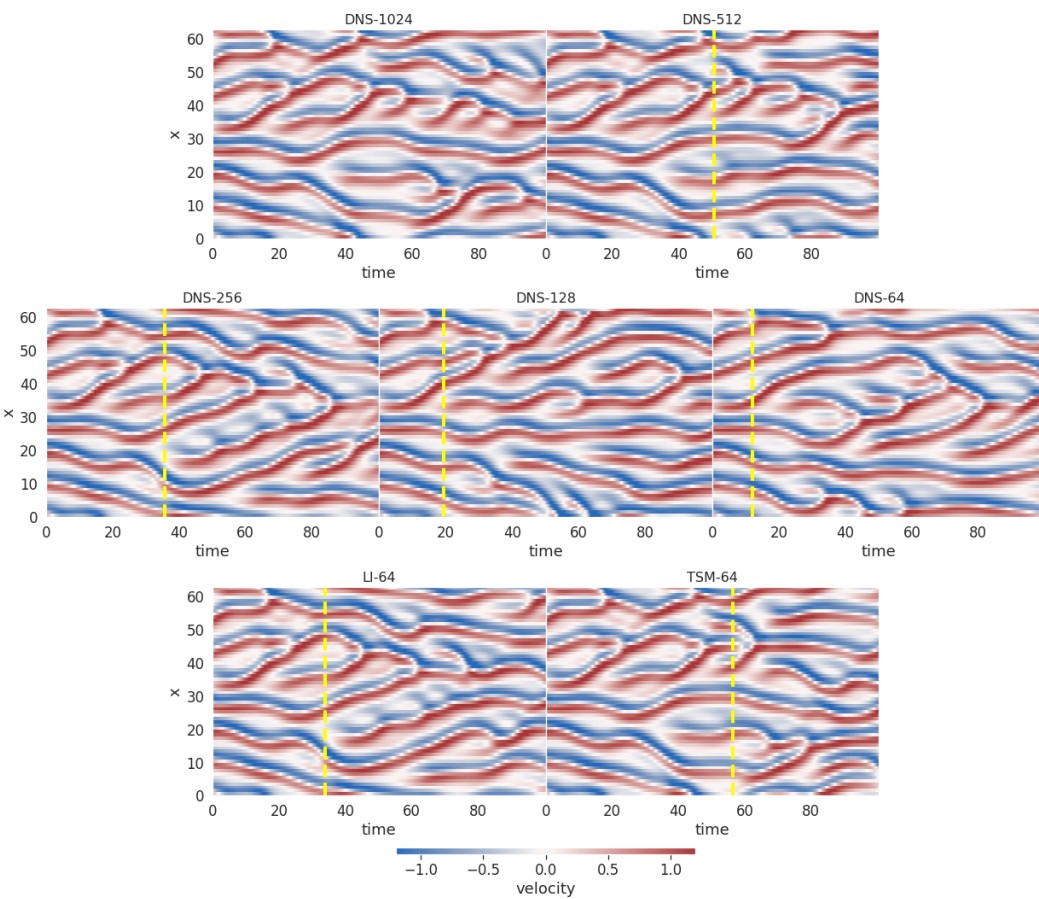

Figure 11: Qualitative comparison of TSM and other baselines on 1D KS equation. The solutions are down-sampled to $64$ grid in the space dimension for comparison. The dashed vertical yellow line denotes the time-step where the Pearson correlation between the model prediction and ground-truth (i.e., DNS-1024) is lower than threshold ($\rho < 0.8$).

Similar to the NS solution, we solve the nonlinear convection term by advecting all velocity components simultaneously, using a high order scheme based on Van-Leer flux limiter or with a learnable interpolator, while the second and forth order diffusion are approximated using a second order central difference approximations. The Fourier spectral method is not used because of the precision issues in JAX FFT and IFFT [6].

We use DNS-1024 as the ground-truth training data, and train TSM solver and LI solver in the $32$ and $64$ down-sampled grids. A time-step of $\Delta t = 1.9635 \times 10^{-2}$ and $\Delta t = 9.81748 \times 10^{-3}$ is used for 32 and 64 grids, respectively. Following Bar-Sinai et al. (2019), the interpolation coefficients of a 6-point stencil are predicted by TSM and LI. For training, we generate $1024$ trajectories of fluid dynamics, each starts with different random initial conditions and simulates with $1024$ resolution for $200.0$ time units (after $80.0$ time unit warmup). We use 16 trajectories for evaluation.

When comparing with other DNS baselines, all solutions are down-sampled to $32$ or $64$ for comparison. Fig. 10 and Fig. 11 show the results of TSM and LI compared with other baselines in the $32$ and $64$ grids, respectively. A quantitative comparison of various solvers under $64$-resolution is also presented in Fig. 12. We can see that TSM can outperforms LI with the same resolution, and outperform DNS with $4\times \sim 8\times$ resolutions.

---

[6] https://github.com/google/jax/issues/2952

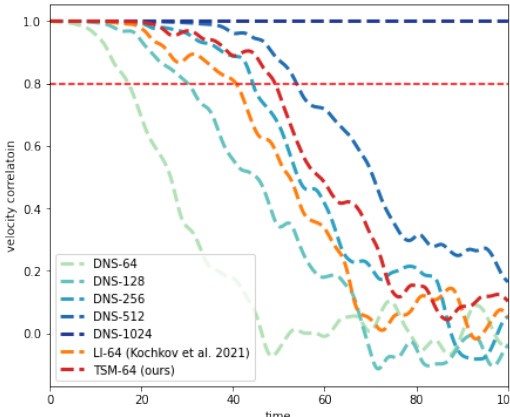

Figure 12: Quantitative comparison of TSM and other baselines on the velocity correlation of 1D KS equation. The solutions are down-sampled to 64 grid in the space dimension for comparison.

## F   GENERALIZATION ON 3D NAVIER-STOKES (NS) EQUATION

To verify the generalization ability of TSM, we further evaluate it on 3D Navier Stokes equations. Recall that the incompressible, constant density Navier Stokes (NS) equation for fluids has a conservation form of:

$$\partial_t \mathbf{u} + \nabla \cdot (\mathbf{u} \otimes \mathbf{u}) = \nu \nabla^2 \mathbf{u} - \frac{1}{\rho} \nabla p + \mathbf{f} \tag{11}$$

We train our method with 3D incompressible Navier-Stokes flow with a linear forcing $\mathbf{f} = 0.05\mathbf{u}$ to avoid the decaying of flows[7]. The viscosity of the fluid is set to $\nu = 6.65 \times 10^{-4}$ and the density $\rho = 1$. For training, we generate 128 trajectories of fluid dynamics, each starts with different random initial conditions and simulates with $128 \times 128 \times 128$ resolution for 10.0 time units (after 80.0 time unit warmup). We use 16 trajectories for evaluation.

Following previous work (Stachenfeld et al., 2021; Takamoto et al.), the ground-truth solution is obtained by simulation on $128 \times 128 \times 128$ grid. According to the Courant–Friedrichs–Lewy (CFD) condition, we have time-step $\Delta = 1.402497 \times 10^{-2}$ on the $32 \times 32 \times 32$ simulation grid. For TSM-$32 \times 32 \times 32$ and LI-$32 \times 32 \times 32$, most of the settings in 3D NS follows our setup for the 2D case, except that the interpolation coefficients is only calculated for a $2 \times 2 \times 2$-point stencil to avoid out-of-memory issues.

When comparing with other DNS baselines, all solutions are down-sampled to $32 \times 32 \times 32$ for comparison. A quantitative comparison of various solvers under $32 \times 32 \times 32$-resolution is presented in Fig. 13. The qualitative results on the planes of $x = 0$, $y = 0$, $z = 0$ are presented in Fig. 14, Fig. 15, and Fig. 16, respectively. We can see that TSM can outperforms LI and DNS with the same resolution, but cannot beat DNS with $2\times$ higher resolution. This is consistent with the results reported in previous work (Stachenfeld et al., 2021).

---

[7]https://github.com/google/jax-cfd/blob/main/jax_cfd/ml/physics_configs/linear_forcing.gin

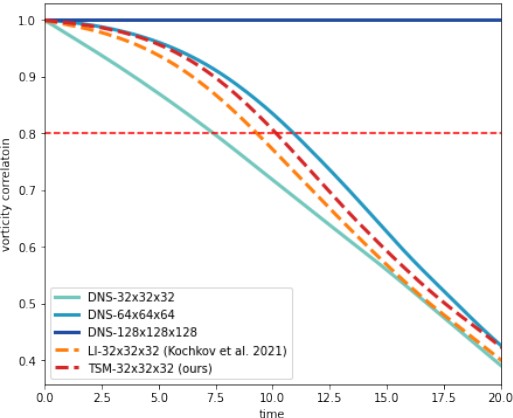

Figure 13: Quantitative comparison of TSM and other baselines on the vorticity correlation (averaged over three directions) of 3D incompressible Navier-Stokes equation.

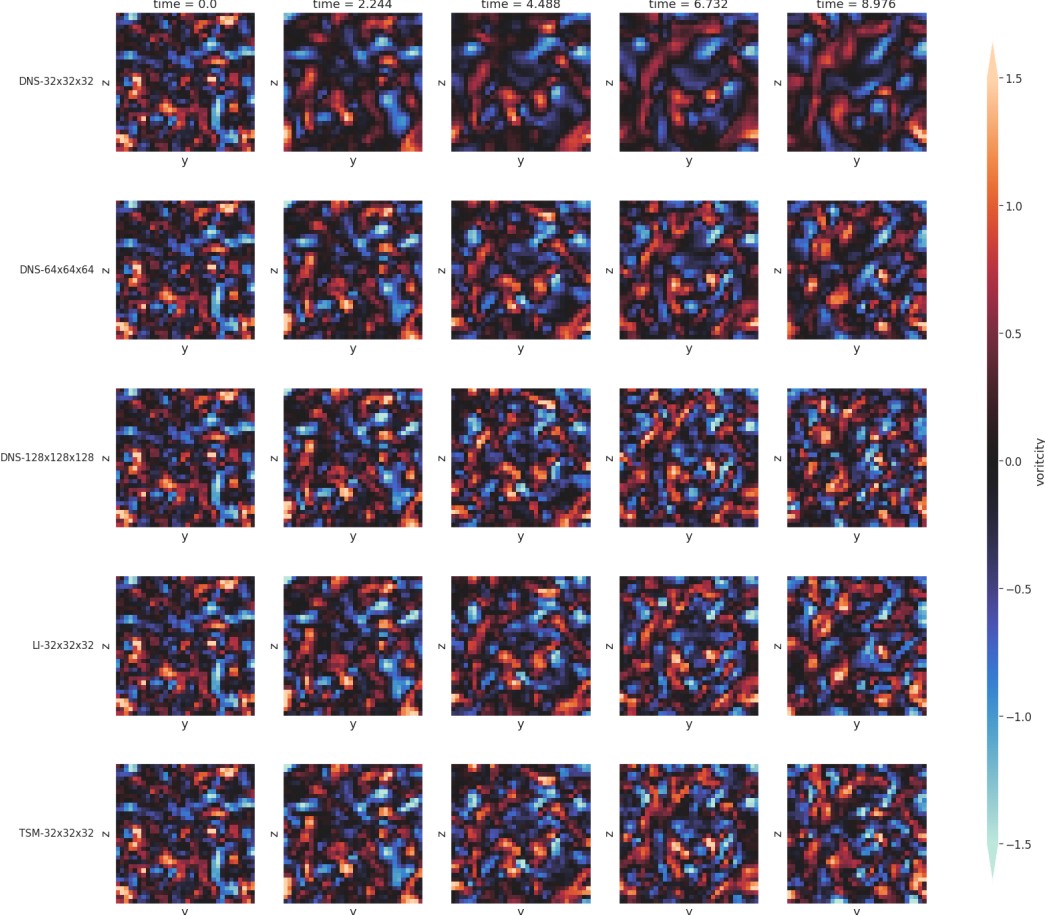

Figure 14: Qualitative 3D incompressible Navier-Stokes equation results of predicted vorticity fields on the plane of $x = 0$.

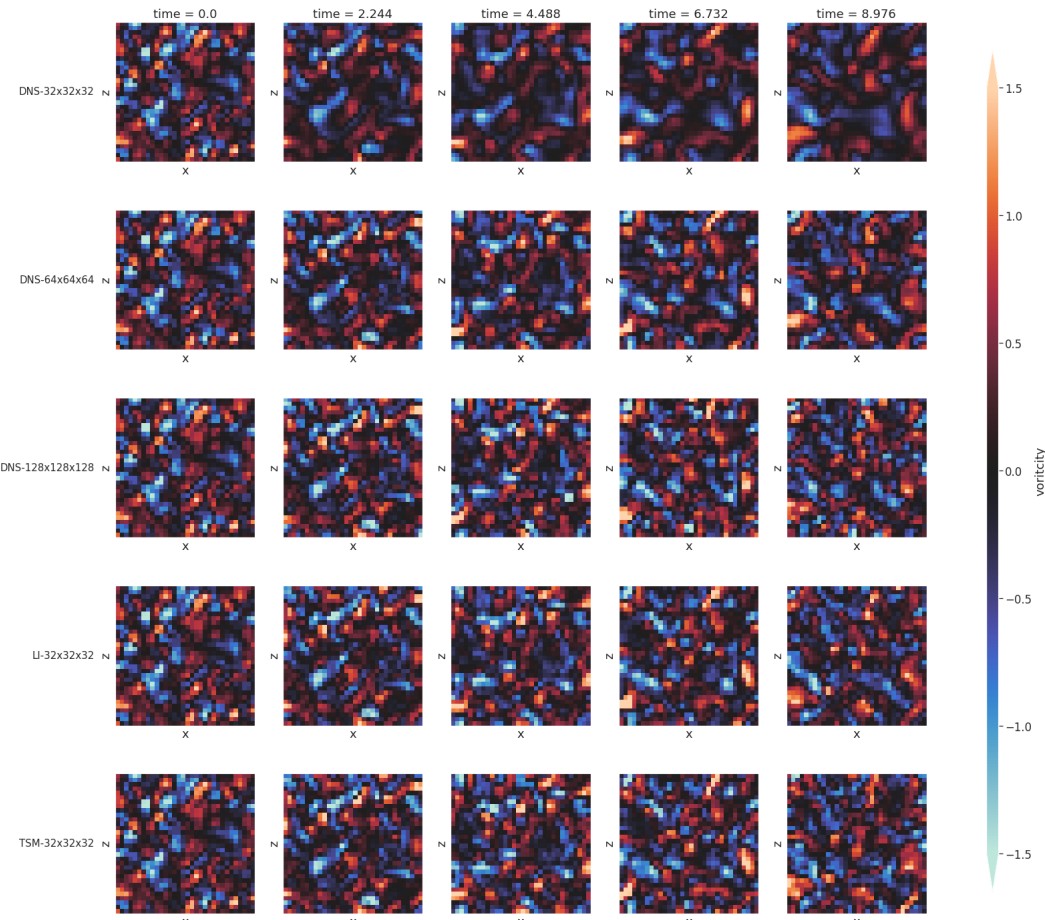

Figure 15: Qualitative 3D incompressible Navier-Stokes equation results of predicted vorticity fields on the plane of $y = 0$.

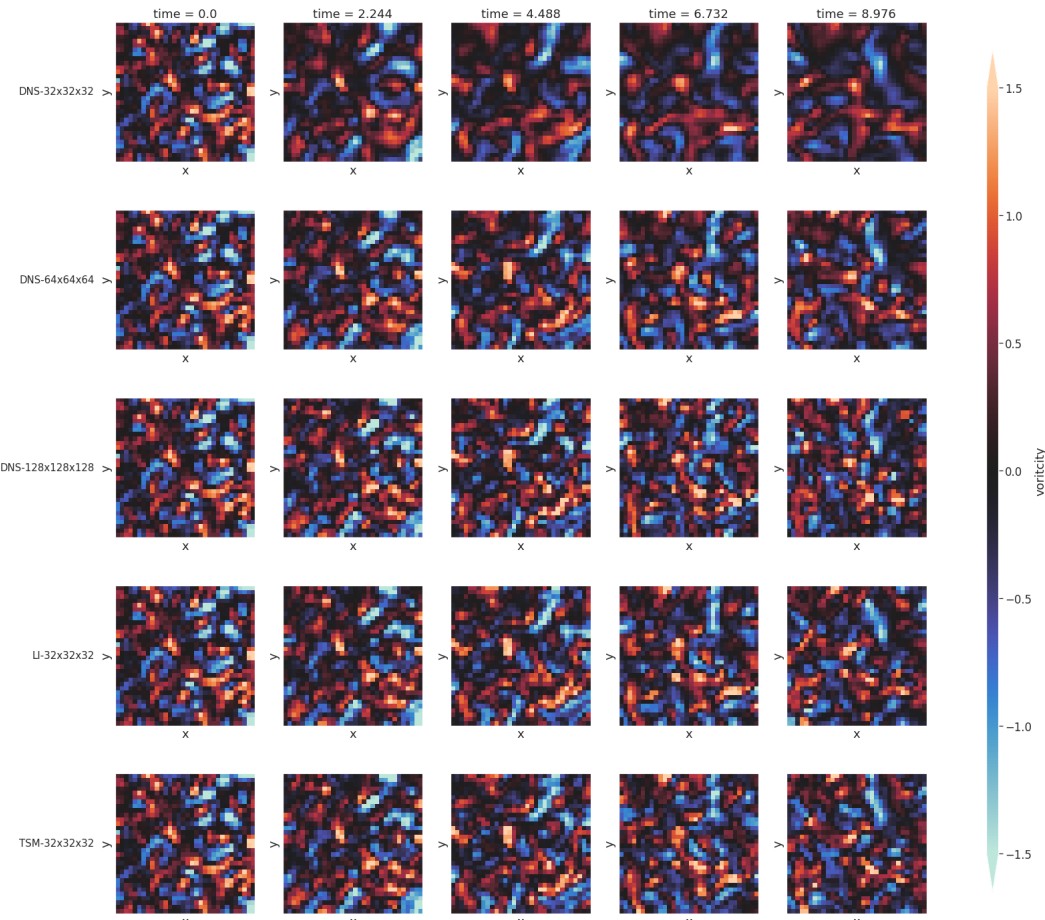

Figure 16: Qualitative 3D incompressible Navier-Stokes equation results of predicted vorticity fields on the plane of $z = 0$.

