# OpenReview forum: "A Neural PDE Solver with Temporal Stencil Modeling"
_ICLR.cc/2023/Conference — Submitted to ICLR 2023_

### Official Review · Reviewer_41X5 · 2022-10-23

**Confidence:** 3
**Correctness:** 3
**Technical Novelty And Significance:** 2
**Empirical Novelty And Significance:** 2
**Recommendation:** 6

**Clarity, Quality, Novelty And Reproducibility:**

Clarity: good.

Quality: fair.

Originality: limited.

Reproducibility: fair.

**Strength And Weaknesses:**

Strength:
The paper addresses an important problem. It is written clearly. The experiment is sound.


Weaknesses:
1. The use a the temporal stencil, and especially the use of HiPPO features is not well motivated. Since this is the most important part of the paper, it can be motivated better, and maybe shorten the background part. Also, it would be nice to give an intuition about the HiPPO features, even if the introduction is in the appendix.

2. The main contribution is a simple combination of the time-series sequence modeling (with the HiPPO features) and SOTA neural PDE solvers, the novelty is limited.

3. It would strengthen the paper, if the method is demonstrated in another (potentially more difficult, larger) dataset, to show the generality and/or scalability of the method.

**Summary Of The Paper:**

This paper introduces a method that combines the strengths of advanced time-series sequence modeling (with the HiPPO features) with neural PDE solvers. In a 2-D Kolmogorov flow, the method shows improved high-correlation duration, compared to baselines with learned correction or raw features.

**Summary Of The Review:**

In summary, the paper introduces a method that improves upon prior SOTA neural solver with the temporal stencil modeling. My main concern is limited novelty and limited experiment evaluation of the paper, as stated in the weakness part of the paper.

---

> ### Author Response · Authors · 2022-11-18
> **Response to Reviewer 41X5**
>
> We thank the reviewer for their time, insightful comments, and questions. We have provided our responses below.
>
> >  it would be nice to give an intuition about the HiPPO features, even if the introduction is in the appendix.
>
> We thank the reviewer for their suggestion. We have added more descriptions to HiPPO in Section 3.2.
>
> > The main contribution is a simple combination of the time-series sequence modeling (with the HiPPO features) and SOTA neural PDE solvers, the novelty is limited.
>
> First, the major difference of TSM from previous neural PDE solvers is the temporal modeling of stencils in a finite volume framework. To the best of our knowledge, previous PDE solvers either only encode the latest timestep for stencil interpolation (e.g., LI or WENO) or encode the temporal trajectories in a pure-ML framework (e.g., FNO).
>
> Second, our paper is the first to apply HiPPO to PDE trajectory modeling and show that HiPPO features can significantly outperform the raw features in three neural PDE frameworks (ML, Learned Corrector framework, and Temporal Stencil Modeling).
>
> Those two factors are indeed the major reason for the success of TSM over existing neural solvers.
>
> > It would strengthen the paper, if the method is demonstrated in another (potentially more difficult, larger) dataset, to show the generality and/or scalability of the method.
>
> We thank the reviewer for the insightful suggestion. In Appendix F (Fig. 10-12) and Appendix G (Fig. 13-16), we report additional results on the 1D Kuramoto–Sivashinsky (KS) equation and 3D incompressible Navier-Stokes equation to further confirm the generality of the proposed method.

---

> > ### Comment · Reviewer_41X5 · 2022-11-26
> > **Thank you for the response**
> >
> > Thank you for the response and the additional experiments. The response mostly addresses my concerns. Thus, I have increased my score.

---

### Official Review · Reviewer_ay4D · 2022-10-24

**Confidence:** 2
**Clarity, Quality, Novelty And Reproducibility:** I have no concerns about the clarity,…
**Correctness:** 4
**Technical Novelty And Significance:** 3
**Empirical Novelty And Significance:** 3
**Recommendation:** 8

**Strength And Weaknesses:**

Strength:

The core idea of the work is clear and simple. The motivation for using temporal feature for super-resolution PDE solution is intuitive and strong. The evaluation of the proposed method is comprehensive.I especially appreciate the generalization test experiment results and the method consistently outperforms existing approaches.


Weakness:

I do not have major concerns about the work except for two minor ones: 1) It’s clear HiPPO feature is used because it’s the SOTA temporal feature but have the authors considered other temporal features like learnable features from RNN and transformer models? 2) There’s a typo in the first paragraph of Section 4.1. Simulated Data-> Simulated Data

**Summary Of The Paper:**

The work proposed a neural PDE solver with temporal stencil modeling. The neural solver incorporate the state of the art temporal HiPPO feature of low-resolution results as inputs for super resolution. The method was evaluated on 2-D Kolmogorov flows and consistently outperform a series of baselines.

**Summary Of The Review:**

The idea of using HiPPO temporal features for neural PDE super-resolution solution is clear and well motivated. Comprehensive experiment results show strong and convincing performance of the proposed methods.

---

> ### Author Response · Authors · 2022-11-18
> **Response to Reviewer ay4D**
>
> > It’s clear HiPPO feature is used because it’s the SOTA temporal feature but have the authors considered other temporal features like learnable features from RNN and transformer models?
>
> We thank the reviewer for the insightful question. To our knowledge, none of the RNN or Transformer models has been used as learnable features for solving PDE problems.
>
> An advantage of the fixed HiPPO feature is that it can be pre-computed and thus enables to avoid the gradient vanishing or out-of-memory problems, which are issues in RNNs/Transformers when modeling long sequences.
>
> Besides, using the parameter-free pre-computed HiPPO recurrence matrix also allows us to make a fair comparison to LI (Kochkov et al., 2021).

---

### Official Review · Reviewer_PBxt · 2022-11-01

**Confidence:** 4
**Correctness:** 2
**Technical Novelty And Significance:** 4
**Empirical Novelty And Significance:** 2
**Recommendation:** 6

**Clarity, Quality, Novelty And Reproducibility:**

The presented work struggles to keep a clear plot of the discussion. While the first third of the paper (up to section 3) is well structured and approachable for a non-CFD reader, the remainder is convoluting domain specific lingo (WENO, HIPPO, FNO, temporal bundle, trajectories) and puts less emphasis on the analysis of the experimental results unfortunately. In addition, many of the plots reported appear made in a rush (e.g. see figure 5: missing units, aspect ratio of plots, multiple legends with and without borders).

The approach appears to bring novelty to field for sure. However, a missing statistical analysis and missing details about the architecture and data set hide important details of the approach (i.e. does the model overfit, how does the loss behave). Finally, the reproducibility statement contains a corrupted URL. By this lack of reproducible code and given the weaknesses indicated earlier, this work may be assumed to NOT be reproducible. The authors could use the rebuttal to fix this flaw easily.

Finally, a central penalty to clarity is the fact, that in the section titled "generalization tests" the approach is found to generalize, while the summary clearly reports that no experiments have been done to check the method with 3D simulations. To me, this is a clear contradiction.

**Strength And Weaknesses:**

The paper is strong in introducing the domain of CFD to a reader from the ML community, i.e. not versed in CFD. The text is easy to follow (despite minor language issues) for an audience versed in partial differential equations (PDEs). The metrics under study are designed in an approachable manner, so that these metrics' interpretation is straight forward. The authors add with indicative visualizations (fig. 2) of the mathematical fields under study. The ML based approach is described in a condensed textual but also visual fashion (fig 1), which is helpful to understand the rational. Towards the end, the authors strive to underpin the trustworthiness of their approach by reporting an ablation study as well as tests for general applicability.

A major weakness of this article is the lack of standard techniques in machine learning. A clear description of the input and output data as well as the network architecture is scattered throughout the paper. It woud be beneficial to have this under 'Experimental Setup'. Moreover, the network architecture used appears to be rather small (6 conv layers with 256 feature channels each). A clear description and comparison to the approach of LI would help.

One more major weakness along this line of thought: a cross-validation (standard technique in ML) could have been undertaken to sample the predictive performance of the network. If done with the competing approach too, this could provide a basis for a statistical treatment of the results and hence increase trustworthiness of any eventual interpretation. Related, the authors claim a "significant" improvement without conducting a statistical analysis of their predictions, i.e. a hypothesis test (or Bayes Factor computation) or estimation of a confidence/credible interval. This severely weakens the interpretations brought forward and raises severe doubts that the findings are significant in the narrow (statistical) sense of the term.


**Summary Of The Paper:**

This paper suggests a Machine Learning (ML) driven method to support solving the Navier-Stokes equation of incompressible flows, a classical problem in Computational Fluid Dynamics (CFD). The submitted approach learns to infer the stencils of advection-diffusion fluids in a Finite Volume Method framework.

The authors introduce the mathematical setting of incompressible constant density Navier Stokes equation for fluids. They then report their ansatz by using embedded HIPPO features and learning a small Convolutional Neural Network (6 layer Unet) to estimate the spatio-temporal stencils. These stencils are used to perform a super-resolution-like task by which a reduced resolution input field (64x64) is regressed to a full-resolution solution of the aforementioned PDE solution (e.g. 2048x2048).

They then describe and show experimental results on low-resolution input data to compare their predictions against the ground truth at high resolution. The comparison is contrasted with one other ML-driven approach taken from [Kochkov et al 2021]. From these results, they infer strong indications that their approach is superior to the existing model by [Kochkov et al 2021].


**Summary Of The Review:**

I congratulate the authors for their submission to ICLR'23. The presented paper demonstrates a novel approach to solving the Navier Stokes PDE in a highly relevant parameter domain (of incompressible flows). This work is hence very relevant for practical concerns and has the potential to push the CFD domain to new PDE or parameter space domains.

The ML-driven TSM approach under study is found to provide strong performance gains beyond one singular competing ML based approach. The evidence in favor of this can be considered weak. I encourage the authors to continue their work and add a more thorough statistical analysis of the predicted fluid fields with their setup. This may include a cross-validation study of the CNN, an estimation of the predicted estimates and a statistical analysis of the obtained confidence intervals.

---

> ### Author Response · Authors · 2022-11-18
> **Response to Reviewer PBxt**
>
> We appreciate the in-depth questions and suggestions given by the reviewer. We have provided our responses below.
>
> > A major weakness of this article is the lack of standard techniques in machine learning. A clear description of the input and output data as well as the network architecture is scattered throughout the paper. It woud be beneficial to have this under 'Experimental Setup'.
>
> We thank the reviewer for the suggestion. We have moved the description of network architectures under the ‘Experimental Setup’ and added a clear description of the input and output.
>
> > Moreover, the network architecture used appears to be rather small (6 conv layers with 256 feature channels each). A clear description and comparison to the approach of LI would help.
>
> The small neural network architectures are chosen for efficiency/latency concerns. The previous work (Kochkov et al., 2021) also used similar small-scale neural network architectures (6/11/25 conv layers with 64 feature channels each). All the neural models (including TSM, LI, LC, and ML) in our comparison use 6 conv layers with 256 feature channels each, which makes it a fair comparison.
>
> > … the authors claim a "significant" improvement without conducting a statistical analysis of their predictions, i.e. a hypothesis test (or Bayes Factor computation) or estimation of a confidence/credible interval…
>
> We thank the reviewer for the suggestion. We present the one-sample T-test results for the differences between TSM-64x64 and other baseline models in Table 4. The differences on all 16 test trajectories are used for each significance test. We evaluate the significant difference for the high-correlation duration time with four thresholds: 0.95, 0.9, 0.8, and 0.7. We find that the differences between TSM and other baselines are consistently statistically significant (e.g., p-value < 2e-02 compared with DNS-1024x1024 and p-value < 4e-3 for LI-64x64).
>
> > does the model overfit
>
> Our both in-domain and out-of-distribution experiments show that TSM consistently outperforms previous ML & physics baselines. Note that the training/validation/test data are generated with different random seeds.
>
> > Finally, the reproducibility statement contains a corrupted URL
>
> The anonymous (not corrupted) URL will be replaced when the paper is accepted.
>
> > Finally, a central penalty to clarity is the fact, that in the section titled "generalization tests" the approach is found to generalize, while the summary clearly reports that no experiments have been done to check the method with 3D simulations. To me, this is a clear contradiction.
>
> First, by “generalization tests”, we mean the out-of-distribution generalization of the neural models (trained with Kolmogorov flows with Re = 1000) on various out-of-distribution simulations. Due to the significant differences between 2D and 3D NS simulations, it is not realistic to train a model with 2D simulations and evaluate it on 3D simulations.
>
> Nonetheless, we report additional results on 1D Kuramoto–Sivashinsky (KS) equation and the 3D incompressible Navier-Stokes equation with linear forcing in Appendix F (Fig. 10-12) and Appendix G (Fig. 13-16), respectively. Our results show that TSM consistently improves the PDE simulation accuracy for 1D, 2D, and 3D cases.

---

### Official Review · Reviewer_EevF · 2022-11-02

**Confidence:** 4
**Correctness:** 3
**Technical Novelty And Significance:** 2
**Empirical Novelty And Significance:** 2
**Recommendation:** 6

**Clarity, Quality, Novelty And Reproducibility:**

In general, the paper is understandable. This method has some novelty that needs more experiments for justification.

**Strength And Weaknesses:**

Strengths :

- TSM-HiPPO encodes raw fluid velocity using the HiPPO, which operates by projecting the previous trajectories as polynomial bases. This approach can well extract features of time-varying fluid dynamic.

- In addition to the ML based model, the authors also experimented with the physics-type model.

- It was confirmed in the paper that TSM-HiPPO has better performance than baselines at various environment not in a single fixed environment.


Weaknesses :

- It is not possible to verify that the proposed model is available for general PDEs. In this paper, the authors verified the performance of the model only for Navier Stokes equation and did not compare it with other PDEs. Additional experiments are needed on various other PDEs that existing models are solving.

- There is a lack of baseline in the neural operator field. In this paper, only Fourier Neural Operators (FNO) are compared as baseline. It is necessary to add models that appeared after FNO as baselines. Additional comparisons with wavelet transform based neural operator(MWNO), transformer based neural operator, and other models that the authors mentioned in Related work are needed.


**Summary Of The Paper:**

The authors proposed TSM that combines time series modeling method HiPPO and neural PDE solver. This proposed method has higher accuracy than existing models in 2-D Navier Stokes equation, and has lower inference latency. The authors generally identified these characteristics in various forcing and Reynolds number environments.

**Summary Of The Review:**

Due to the following concerns, I think the paper is not at the acceptance level yet:

- Did the authors compare other basic deep learning models besides CNN?

- It seems necessary to compare it with other models that perform the Super-resolution task in the PDE Problem. Neural Operator models also perform super-resolution tasks. It is necessary to compare.

- TSM does not seem to have significantly changed from the existing HiPPO and Neuarl PDE solver models. Can you explain this in more detail?

- I wonder if using 64 x 64 grid for all learnable solvers in Table 2 is a fair comparison. How does the performance of the models change for different resolutions?

However, I think this paper adds some value to the physics-informed machine-learning community. During the rebuttal period, I recommend they address as many issues as possible and I will reconsider my evaluation in a positive way.

---

> ### Author Response · Authors · 2022-11-18
> **Response to Reviewer EevF**
>
> We thank the reviewer for their time, insightful comments, and questions. We have provided our responses below.
>
> > Additional experiments are needed on various other PDEs that existing models are solving.
>
> We have added additional experiments on the 1D Kuramoto–Sivashinsky equation and 3D incompressible Navier-Stokes equation in Appendix F (Fig. 10-12) and Appendix G (Fig. 13-16), respectively. Our results show that TSM consistently improves the PDE simulation accuracy for 1D, 2D, and 3D cases.
>
> > There is a lack of baseline in the neural operator field.
>
> We present additional experimental results of the multiwavelet transform based neural operator (MWNO) in Table 1 as a baseline for comparison. The main finding from the additional experiments is that both LI and TSM still outperform the recently proposed neural operators.
>
> > Did the authors compare other basic deep learning models besides CNN?
>
> For pure deep learning models (without the physics module), beyond HiPPO-CNN and raw-CNN, we also compare with FNO and WMNO, as shown in Table 1.
>
> > It seems necessary to compare it with other models that perform the Super-resolution task in the PDE Problem. Neural Operator models also perform super-resolution tasks. It is necessary to compare.
>
> In general, neural operator models (including FNO and WMNO) cannot perform super-resolution tasks for PDE. On the other hand, our super-resolution model uses the de-convolution operation, which is resolution-dependent. Standard neural operator models (such as FNO) cannot perform super-resolution unless the local linear transform is replaced with a de-convolution operation.
>
> > TSM does not seem to have significantly changed from the existing HiPPO and Neuarl PDE solver models. Can you explain this in more detail?
>
> First, the major difference of TSM from previous neural PDE solvers is the temporal modeling of stencils in a finite volume framework. To the best of our knowledge, previous PDE solvers either only encode the latest timestep for stencil interpolation (e.g., LI or WENO) or encode the temporal trajectories in a pure-ML framework (e.g., FNO).
>
> Second, our paper is the first to apply HiPPO to PDE trajectory modeling and show that HiPPO features can significantly outperform the raw features in three neural PDE frameworks (ML, Learned Corrector framework, and Temporal Stencil Modeling).
>
> > I wonder if using 64 x 64 grid for all learnable solvers in Table 2 is a fair comparison. How does the performance of the models change for different resolutions?
>
> We have already provided the comparative evaluation results on 32 x 32 and 16 x 16 grids in Appendix E, where TSM still significantly outperforms LI in the same resolution and DNS in 4-8x larger resolutions.

---

> ### Author Response · Authors · 2022-12-05
> **Have our responses addressed your concerns?**
>
> Dear Reviewer EevF,
>
> We hope the responses and the revised manuscript can address your concerns and eagerly look forward to receiving your feedback.
>
> Thank you!

---

### Author Response · Authors · 2022-11-18
**General Response**

We thank all the reviewers for their precious time and insightful comments. We appreciate that the reviewers recognize our work as novel (Reviewer EevF, Reviewer PBxt, Reviewer ay4D), written clearly (Reviewer 41X5, Reviewer ay4D), and having sound evaluation (Reviewer EevF, Reviewer 41X5). To improve the paper quality, we respond to the reviewers’ comments by making the following major revisions to the paper:

1. We have moved the description of network architectures under the ‘Experimental Setup’ and added a clear description of the input and output.
2. We present the one-sample T-test results for the differences between TSM-64x64 and other baseline models in Table 4. The differences on all 16 test trajectories are used for each significance test. We evaluate the significant difference for the high-correlation duration time with four thresholds: 0.95, 0.9, 0.8, and 0.7. We find that the differences between TSM and other baselines are consistently statistically significant (e.g., p-value < 2e-02 compared with DNS-1024x1024 and p-value < 4e-3 for LI-64x64).
3. We present additional experimental results of the multiwavelet transform based neural operator (MWNO) in Table 1 as a baseline for comparison. The main finding from the additional experiments is that both LI and TSM still outperform the recently proposed neural operators.
4. We have added additional experiments on the 1D Kuramoto–Sivashinsky equation and 3D incompressible Navier-Stokes equation in Appendix F (Fig. 10-12) and Appendix G (Fig. 13-16), respectively. Our results show that TSM consistently improves the PDE simulation accuracy for 1D, 2D, and 3D cases.

---

### Decision · Program_Chairs · 2023-01-20

**Decision:**

Reject

**Justification For Why Not Higher Score:**

The original contributions remains incremental w.r.t. previous work and the paper would benefit from an improved technical description more adapted to a ML audience.

**Justification For Why Not Lower Score:**

N/A

**Metareview: Summary, Strengths And Weaknesses:**

The paper introduces a hybrid model combining ML modules and a PDE solver for solving an incompressible Navier-Stokes equation. The objective is to solve turbulent problems at a reduced cost. The ML component is trained to predict coefficients involved in the computation of spatial derivative approximations (the stencil coefficients) as used in finite volume solvers. The coefficients then depend on the space and time through their dependence on the field values of neighboring cells used in the approximation. These approximations are used to compute the flux of the flow involved in the equation. This allows the model to infer solutions at a high-resolution at the cost of low-resolution simulations. Experiments are performed on 2D incompressible NS and on 1 D Kuramoto-Sivashinsky equation. Qualitative results are also shown for a 3D NS equation.

The paper addresses the important problem of accelerating computational fluid dynamics simulations. The work presented here closely follows the framework and the experimental methodology introduced in previous work, ref (Kochkov 2021) in the paper. The main novelty consists in extending this framework by considering a series of successive velocity field frames for predicting the stencil coefficients instead of considering a single frame, i.e. the velocity field at one time only (Kochkov 2021). Two strategies are proposed for integrating the frame sequence, one consists in considering the raw series of previous states as a space-time tensor, and the other one relies on an encoding of this sequence through a recent method introduced for time series. The experiments show that the proposed idea improves over previous work.

During the rebuttal period, the authors improved the paper presentation and added new experiments as asked for by the reviewers, which strengthens the paper. The novelty of the contribution remains however incremental w.r.t. previous work (Kochkov 2021) – the main difference being the temporal state sequence used for modeling the stencils instead of a single system state. The performance improvements are considered here relatively modest. Besides the description and analysis of the experiments could still be largely improved, especially for making the paper more accessible to a ML audience.